# From Language to Action Streams: Bridging LLM Autoregression for Long-Horizon Robot Action Prediction

## Abstract

Vision-Language-Action (VLA) models is a transformative paradigm for robotic control, leveraging pre-trained vision-language models (VLMs) to directly translate natural language instructions and visual observations into low-level actions. The prominent idea of "Action-as-Language" discretizes action spaces into tokens for large language models (LLMs), reframing action prediction as a standard sequential language generation task. However, current implementations underutilize the LLM's full generation potential, confining action prediction to fixed-length, single-step token sequences and limiting the policy's generation horizon. To overcome this limitation, we propose the **Action Stream** paradigm, which customizes LLM training and inference recipes to VLAs, enabling the generation of extended chains of action tokens and facilitating implicit long-horizon generation with task performance improvements. For training action streams, we propose a two-phase approach: Long-horizon Behavior Cloning (L-BC) and Step-wise Action Alignment (S-AA). L-BC enables VLA models to generate coherent multi-step action sequences, while S-AA mitigates exposure bias during sequential inference, creating a framework that enables long-horizon generation while reducing error accumulation. During deployment, decoding strategies from language generation can be successfully transferred to action streams, enabling efficient solution search and task performance improvements. Through extensive evaluations on the simulation benchmark and real-world robotic setups, we demonstrate that the Action Stream paradigm achieves improved task performance when extending the generation horizon, representing a significant step toward unified vision-language-action modeling.

## 1 Introduction

Vision-Language-Action (VLA) models have emerged as the leading approach for general robot control, providing end-to-end systems that translate natural language instructions and visual inputs directly into executable robotic actions (Brohan et al., 2022; 2024; Belkhale et al., 2024; Black et al.; Kim et al., 2025). By adapting pre-trained Vision-Language Models (VLMs) for action prediction (Wang et al., 2025; Zhai et al., 2025; Li et al., 2023), VLA models leverage both the language understanding capabilities of LLMs and visual perception of vision networks to ground abstract instructions in concrete robotic actions.

To endow VLMs with the ability of action prediction, the academic community has explored two main paradigms. The first is the "Module Grafting" paradigm, which grafts an independent action prediction module (such as an MLP regression head (Jang et al., 2022) or a diffusion decoder (Chi et al., 2023)) onto the VLM's feature representation. While effective, this introduces architectural complexity by requiring the integration of heterogeneous modules and necessitates specialized training strategies (Liu et al., 2025; Kim et al., 2025; Bu et al., 2025).

Unlike the modular approach, the second paradigm proposes a more elegant unified perspective by rewriting the output space of VLMs, with the core idea summarized as "Action-as-Language." Represented by RT-2 (Brohan et al., 2024) and OpenVLA (Kim et al., 2024), this paradigm is realized through the technique of "Action Tokenization": it discretizes the continuous robotic action and then

directly maps each discrete action to existing tokens in the LLM's native vocabulary. This design repurposes the LLM's output space, making action generation identical to language generation and enabling VLA models to use standard language model training and inference methods, significantly enhancing architectural simplicity and fine-tuning efficiency.

However, the Action-as-Language paradigm is critically limited by its failure to leverage the LLM's full generation capabilities. Current models like OpenVLA (Kim et al., 2024) predict actions one step at a time with short, fixed-length token sequences. This approach, while effective for reactive control, fails to leverage the LLM's strengths for generating coherent, extended sequences, limiting the policy's horizon. It misses opportunities for implicit long-horizon planning and temporal dependencies that could emerge from generating multi-step actions autoregressively (Liu et al., 2024; Zhao et al., 2023), representing a significant gap between the policy's potential and its current application.

To address this limitation, we introduce the "Action Stream" paradigm, which incorporates LLM training and inference recipes into VLA and transforms robotic actions into a chain of action tokens, enabling policies to generate multi-step plans through autoregressive generation. To realize this, we propose two specialized fine-tuning phases that customize established LLM training recipes (Ouyang et al., 2022; Shengyu et al., 2023; Rafailov et al., 2023) for VLA models. The initial phase, Long-horizon Behavior Cloning (L-BC), adapts the base VLA model to generate coherent, extended action token sequences through offline imitation learning on restructured expert demonstrations. This phase transforms the single-step prediction policy into one capable of generating coherent action streams for long-horizon generation. In the second phase, we implement Step-wise Action Alignment (S-AA), an online alignment technique that addresses the exposure bias problem from the first phase. During the first behavior cloning phase, the model is trained with teacher forcing using ground-truth inputs for next action prediction, but must rely on its own predictions during inference, creating a distribution mismatch that leads to compounding errors in long-horizon execution (Bengio et al., 2015; Wulfmeier et al., 2024; Cundy & Ermon, 2023; Bachmann & Nagarajan, 2024). S-AA addresses this challenge by using preference rewards to identify potential error steps in online long-horizon generation and align them with expert demonstrations. Our experiments on the simulation benchmark and real-world robotic setups show that L-BC alone successfully extends the policy's generation horizon, enabling longer-horizon task completion. With the addition of S-AA tuning, we achieve even greater performance gains, maintaining extended horizons while significantly improving task success rates. This two-phase approach creates robust policies that extend the planning horizon while achieving improved performance, outperforming conventional single-step approaches across various manipulation tasks.

Additionally, during policy deployment, we adapt inference-time computation paradigms to action streams. Unlike traditional fixed sampling methods, we apply diverse sampling and search strategies from LLM generation recipes, which significantly improve task performance. These improvements demonstrate that the action stream paradigm creates a richer solution space that can be effectively explored using established LLM decoding techniques, further enhancing robotic manipulation capabilities.

## 2 PRELIMINARY

### 2.1 VISION-LANGUAGE-ACTION (VLA) MODELS

Our work builds upon OpenVLA (Kim et al., 2024), a powerful open-source VLA model. We briefly introduce its core components below.

**Architecture** OpenVLA is built on a VLM framework, combining SigLIP (Zhai et al., 2023) and DINOV2 (Oquab et al., 2023) visual encoders with a Llama-2-7B LLM (Touvron et al., 2023) backbone for generation. At each time step, the policy takes an RGB image and language instruction as input. The image is encoded into embedding tokens while the instruction is tokenized via the Llama-2 tokenizer. These token sequences are concatenated and processed by the causal transformer decoder to generate outputs.

**Action Tokenization and Vocabulary Remapping** To enable an LLM to generate physical commands, OpenVLA bridges the continuous robot action space with the LLM's text generation capabilities through discretization. The 7-dimensional action vector (6-DoF end-effector displacement

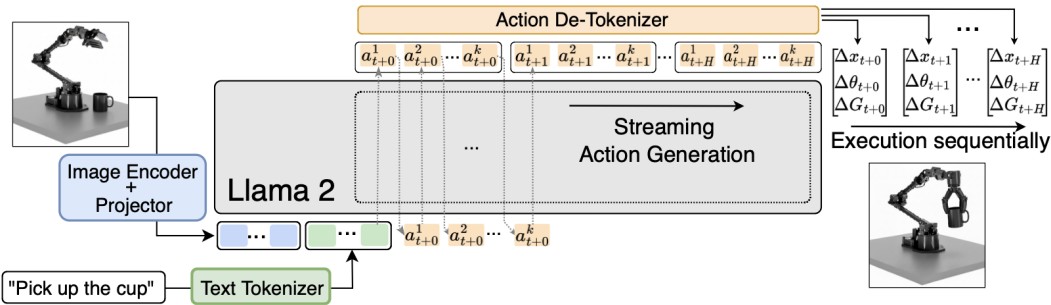

Figure 1: The illustration of the proposed Action Stream Paradigm. We transfer the text generation paradigm to long-horizon action generation, enabling the policy to generate coherent multi-step action sequences from current states, similar to how LLMs produce coherent text.

plus gripper state) is uniformly discretized into 256 bins per dimension following (Brohan et al., 2022; 2024), representing each action as seven integer indices. These indices are mapped to the 256 least-frequently used tokens in the Llama-2 vocabulary, transforming action prediction into a standard text generation task where the policy autoregressively predicts action tokens just as it would predict words in a sentence.

## 2.2 DIRECT PREFERENCE OPTIMIZATION (DPO)

Direct Preference Optimization (DPO) (Rafailov et al., 2023) is a powerful alignment paradigm that aligns LLMs with human preferences using static preference datasets, eliminating the need for explicit reward models or complex online reinforcement learning as required in RLHF (Ouyang et al., 2022).

DPO is based on the Bradley-Terry preference model (Sun et al., 2024), where the probability of winner response $y_w$ over loser $y_l$ given prompt $x$ is $\sigma(r(y_w|x) - r(y_l|x))$, with $r(\cdot|x)$ being a latent reward function. Importantly, DPO demonstrates that the policy $\pi_\theta$ implicitly defines this reward as:

$$r_\theta(y \mid x) = \beta \log \frac{\pi_\theta(y \mid x)}{\pi_{\text{ref}}(y \mid x)} + f(x),$$

where $\pi_{\text{ref}}$ is a reference policy, $\beta$ controls preference strength, and $f(x)$ is a normalization constant.

To optimize the policy, DPO uses a preference dataset of triplets $(x, y_w, y_l)$ and the alignment objective is equivalent to minimizing a the following loss:

$$\mathcal{L}_{\text{DPO}}(\pi_\theta; \pi_{\text{ref}}) = -\mathbb{E}_{(x,y_w,y_l)\sim\mathcal{D}} \left[ \log \sigma \left( \beta \log \frac{\pi_\theta(y_w \mid x)}{\pi_{\text{ref}}(y_w \mid x)} - \beta \log \frac{\pi_\theta(y_l \mid x)}{\pi_{\text{ref}}(y_l \mid x)} \right) \right].$$

This loss function continuously aligns the policy with the winner responses by increasing their likelihood while decreasing that of the loser responses, effectively optimizing the policy to match the preferences encoded in the dataset.

## 3 METHOD

### 3.1 PROBLEM FORMULATION

The robot control task is formulated as a sequential decision-making problem within a Markov Decision Process (MDP) framework. The state at any given time step $t$ is defined as $s_t = (v_t, l)$, where $v_t$ is the current visual observation and $l$ is the high-level language instruction, which remains constant throughout the task episode.

The action space is defined by the "action tokenization" scheme of OpenVLA. A single, complete action at time step $t$, which we term an **action unit** and denote as $\mathbf{a}_t$, is represented by a sequence of $K$ discrete tokens:

$$\mathbf{a}_t = (a_t^1, a_t^2, \ldots, a_t^K) \tag{1}$$

where $K = 7$ for a 7-DoF action (6-DoF end-effector displacement + 1 gripper state), and each $a_t^k$ is an integer index corresponding to a specific token in the VLM's vocabulary.

**The Conventional Single-Step Prediction Paradigm**  The original OpenVLA policy, which we denote as $\pi_{\text{base}}$, operates as a closed-loop, single-step predictor. At each time step $t$, it generates a single action unit $\mathbf{a}_t$ conditioned on the current state $s_t$ through an autoregressive process over the $K$ constituent tokens:

$$\pi_{\text{base}}(\mathbf{a}_t|s_t) = \prod_{k=1}^{K} P(a_t^k|s_t, a_t^{<k}) \tag{2}$$

This "one-step-at-a-time" paradigm, while effective for reactive control, inherently limits the policy's planning horizon and fails to exploit the LLM backbone's intrinsic capability for generating long, coherent token sequences.

**The Proposed Action Stream Paradigm.**  To address this limitation, we reformulate the task from single-step prediction to multi-step **Action Stream** generation. Our goal is to train a policy $\pi_\theta$ that, given a single observation $s_t$, can directly generate a sequence of future action units $(\mathbf{a}_t, \mathbf{a}_{t+1}, \dots)$ as an open-loop plan. Formally, we define an Action Stream of horizon $H$ starting from time $t$ as:

$$\mathcal{A}_t^H = (\mathbf{a}_t, \mathbf{a}_{t+1}, \dots, \mathbf{a}_{t+H-1}) \tag{3}$$

The generation process follows a two-level autoregressive structure: as shown in Figure 1, at the outer level, the policy sequentially predicts each future action unit, and at the inner level, each action unit is generated token by token. This results in the following full conditional probability:

$$P(\mathcal{A}_t^H|s_t) = \prod_{h=0}^{H-1} P(\mathbf{a}_{t+h}|s_t, \mathcal{A}_t^{<h}) = \prod_{h=0}^{H-1} \left( \prod_{k=1}^{K} P(a_{t+h}^k|s_t, \mathcal{A}_t^{<h}, a_{t+h}^{<k}) \right) \tag{4}$$

where $\mathcal{A}_t^{<h}$ denotes the sequence of action units $(\mathbf{a}_t, \dots, \mathbf{a}_{t+h-1})$ generated for previous time steps within the stream.

We briefly introduce that this objective will be achieved via a two-stage framework: **Stage 1:** Long-Horizon Behavior Cloning (L-BC), and **Stage 2:** Step-wise Action Alignment (S-AA).

### 3.2 Stage 1: Offline Long-Horizon Behavior Cloning

The pre-trained OpenVLA model, $\pi_{\text{base}}$, is limited to generating only 7 tokens per action unit. To enable long-horizon generation capability, we train the policy to imitate expert action streams through Supervised Fine-Tuning (SFT).

**Data Reformatting for Stream Imitation.**  We restructure the standard expert demonstration dataset from individual $(s_t, \mathbf{a}_t)$ pairs into a format suitable for long action stream modeling. For each state $s_t$ in an expert trajectory, we construct a ground-truth "Expert Action Stream" $\mathcal{A}_{t,E}^H = (\mathbf{a}_t, \mathbf{a}_{t+1}, \dots, \mathbf{a}_{t+H-1})$ by concatenating the subsequent $H$ action units. We insert a special separator token `[;]` between consecutive units to delineate action unit boundaries. The final target sequence $A_t^E$ is a flat sequence of tokens:

$$A_{t,E}^H = \text{concat}(\mathbf{a}_t, \texttt{[;]}, \mathbf{a}_{t+1}, \texttt{[;]}, \dots, \mathbf{a}_{t+H-1}, \texttt{[;]}) \tag{5}$$

This reformatting process transforms the original dataset $\mathcal{D} = \{(s_t, \mathbf{a}_t)\}$ into a new instruction-following dataset $\mathcal{D}_{\text{stream}} = \{(s_t, A_{t,E}^H)\}$.

**Training Objective.**  The SFT stage closely mirrors the instruction fine-tuning process in LLMs. The state $s_t$ serves as the "instruction", and the expert action stream $A_{t,E}^H$ serves as the ground-truth "response" (Shengyu et al., 2023). We train the policy $\pi_\theta$ using supervised fine-tuning to maximize the log-likelihood of generating the expert action stream conditioned on a given state, minimizing the standard negative log-likelihood loss:

$$\mathcal{L}_{\text{SFT}}(\theta) = -\mathbb{E}_{(s_t, A_{t,E}^H) \sim \mathcal{D}_{\text{stream}}} \left[ \log \pi_\theta(A_{t,E}^H|s_t) \right] \tag{6}$$

where the log-probability is decomposed autoregressively over the tokens of the target sequence:

$$\log \pi_\theta(A_{t,E}^H|s_t) = \sum_{j=1}^{|A_{t,E}^H|} \log P_\theta(a_{t,E}^j|s_t, a_{t,E}^{<j})$$

Here, $a_{t,E}^j$ is the $j$-th token in the flattened sequence $A_{t,E}^H$. By fine-tuning on these structured pairs, we explicitly teach the model to interpret a state as a request for a coherent, multi-step plan, effectively reformatting its output behavior. The resulting policy from this stage is denoted as $\pi_{\text{SFT}}$.

### 3.3 STAGE 2: ONLINE STEP-WISE ACTION ALIGNMENT

**Challenge**  The L-BC stage enables $\pi_{\text{SFT}}$ to generate long-form action sequences. However, it relies on teacher forcing during training, where the model predicts the next token conditioned on ground-truth expert tokens (Wulfmeier et al., 2024; Williams & Zipser, 1989). This creates a mismatch with inference conditions, leading to exposure bias: during deployment, the policy must condition on its own generated tokens, potentially resulting in errors (Bengio et al., 2015; Bachmann & Nagarajan, 2024). These errors can cause irreversible environmental changes, making recovery difficult and causing action sequences to diverge, ultimately leading to task failure.

**Methodology**  To address this challenge, we introduce an online exploration and alignment paradigm based on Direct Preference Optimization (DPO) that moves beyond passive learning.

Given a state $s_t$, we first allow the policy $\pi_\theta$ to actively explore and generate its own action stream step by step: $\mathcal{A}_{t,\pi}^H = (\mathbf{a}_{t,\pi}, \ldots, \mathbf{a}_{t+H-1,\pi})$. We then retrieve the corresponding expert's action stream: $\mathcal{A}_{t,E}^H = (\mathbf{a}_{t,E}, \ldots, \mathbf{a}_{t+H-1,E})$. To identify the first position where deviation occurs in the policy's rollout, we leverage the implicit preference reward derived from the DPO formulation.

$$R(\mathbf{a}|c;\theta,\pi_{\text{ref}}) = \beta \log \frac{\pi_\theta(\mathbf{a}|c)}{\pi_{\text{ref}}(\mathbf{a}|c)} \tag{7}$$

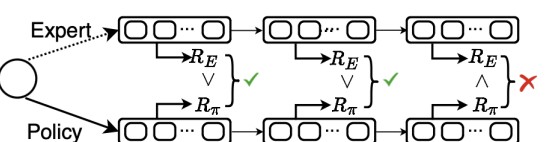

Figure 2: Illustration of our proposed S-AA: given a state and expert action stream, the policy explores multi-step actions, then calculates preference reward at each step to identify the first divergence and applies DPO loss at that step to align with the expert.

Here, the reference policy $\pi_{\text{ref}}$ is initialized from the SFT policy $\pi_{\text{SFT}}$. This reward quantifies how the policy's preference for an action has changed relative to the initial expert-imitating policy $\pi_{\text{SFT}}$, with higher values indicating stronger preference compared to its initial version. Then we compare the preference divergence between the policy's generated action and the expert's action. We define an deviation at step $h^*$ when the policy's action $\mathbf{a}_{\pi,h^*}$ has a higher preference reward than the expert's action $\mathbf{a}_{E,h^*}$ at the same context, formally expressed as $R(\mathbf{a}_{\pi,h^*}|c_{h^*}) > R(\mathbf{a}_{E,h^*}|c_{h^*})$. This indicates that the policy has developed an overconfidence in its own potentially suboptimal action compared to the expert demonstration. Once the first deviation is identified, we apply the DPO loss exclusively at this specific step $h^*$. The objective is to maximize the preference for the expert's action $\mathbf{a}_{E,h^*}$ over the policy's action $\mathbf{a}_{\pi,h^*}$ at that specific context $c_{h^*}$, with the loss function defined as:

$$\mathcal{L} = -\mathbb{E}_{s_t \sim \mathcal{D}}\left[\mathbb{I}(h^* \text{ exists}) \cdot \log \sigma\left(\beta \log \frac{\pi_\theta(\mathbf{a}_{E,h^*}|c_{h^*})}{\pi_{\text{ref}}(\mathbf{a}_{E,h^*}|c_{h^*})} - \beta \log \frac{\pi_\theta(\mathbf{a}_{\pi,h^*}|c_{h^*})}{\pi_{\text{ref}}(\mathbf{a}_{\pi,h^*}|c_{h^*})}\right)\right] \tag{8}$$

where $\mathbb{I}(\cdot)$ is the indicator function, ensuring the loss is only active when a first mistake is found, and the reference policy $\pi_{\text{ref}}$ remains the frozen SFT policy $\pi_{\text{SFT}}$.

By optimizing this objective, we are not just performing imitation with teacher forcing. Instead, we introduce online exploration that actively identifies and corrects potential error actions in real-time. This approach pinpoints the precise origin of behavioral drift, specifically the very first action that would lead the policy down a suboptimal path, and immediately aligns it with expert behavior, thereby mitigating the accumulation of errors in long-sequence generation.

Table 1: Success rates across different generation horizons, task suites, and training stages.

| | H | Spatial | Object | Goal | Long | Average |
|---|---|---|---|---|---|---|
| OpenVLA | 1 | 84.4±0.9% | 86.6±0.8% | 77.2±1.0% | 53.7±1.3% | 75.5±1.0% |
| L-BC | ×2 | 83.6±0.8% | 85.6±0.9% | 78.4±0.9% | 53.4±1.2% | 75.2±0.9% |
| | ×4 | 79.4±1.0% | 81.4±0.7% | 75.4±0.9% | 52.4±1.1% | 72.2±0.9% |
| | ×6 | 73.2±0.9% | 76.2±0.9% | 67.8±0.8% | 48.0±1.0% | 66.3±0.9% |
| | ×8 | 64.6±1.0% | 71.2±1.1% | 58.2±0.7% | 39.6±0.9% | 58.4±0.9% |
| | ×10 | 53.0±1.2% | 62.8±1.4% | 48.8±1.3% | 27.2±1.2% | 47.9±1.3% |
| L-BC+S-AA | ×2 | **85.4±0.8%** | **89.0±0.7%** | **80.0±0.9%** | **57.6±1.1%** | **78.0±0.9%** |
| | ×4 | 82.0±0.7% | 84.0±0.7% | 76.8±0.9% | **54.2±1.0%** | 74.2±0.8% |
| | ×6 | 75.4±0.6% | 78.0±0.6% | 70.0±0.8% | 47.6±0.9% | 67.8±0.7% |
| | ×8 | 65.8±0.5% | 73.6±0.5% | 60.6±0.7% | 41.2±0.8% | 60.3±0.6% |
| | ×10 | 55.6±0.4% | 66.0±0.4% | 52.0±0.6% | 29.6±0.7% | 50.8±0.5% |

## 4 EXPERIMENTS

### 4.1 SETUP

We evaluate our method on the LIBERO dataset (Liu et al., 2023), which consists of four task suites (each containing 10 tasks with 50 human-teleoperated demonstrations): LIBERO-Spatial , LIBERO-Object , LIBERO-Goal, and LIBERO-Long. To validate our approach beyond simulation, we conduct experiments on a **Kinova Jaco2** 6-DoF robotic arm with a parallel-jaw gripper. Four tabletop manipulation tasks are designed, covering pick-and-place and wiping motions with diverse objects. Details can be found in the Appendix A.2 and A.5.

We use OpenVLA as our backbone with LoRA (Hu et al., 2022) fine-tuning. For the first stage (offline action stream imitation), we follow standard SFT paradigm commonly used for LLMs on an A6000 GPU with batch size 4, content length 512, learning rate 5e-4, and train for 50,000 steps. In the second stage (online alignment), we use training trajectories as expert demonstrations with $\beta = 0.1$ and learning rate 0.0005, continuing to fine-tune the same LoRA parameters. Additional details are in the Appendix A.4.

### 4.2 ABLATION ON ACTION STREAM HORIZON LENGTH

This experiment investigates the core proposition of our work: that the Action Stream policy can extend the action generation horizon similar to how LLMs generate coherent text, extended generation horizons. We analyze the impact of varying the horizon length $H$ on two different settings: **L-BC only**: The model trained only with Long-Horizon Behavior Cloning (Stage 1), to assess the benefits of simply enabling long-sequence generation. **L-BC + S-AA**: The full model, fine-tuned with our proposed Step-wise Action Alignment (Stage 2), to demonstrate its ability to mitigate compounding errors.

**Results.** Table 1 shows how horizon length $H$ affects task success rates across training regimes. For L-BC only models, performance shows minimal decline when $H$ increases from 1 to 2 (0.3% difference) and modest reduction at $H = 4$ (3.1% decrease). However, performance drops sharply at longer horizons: $H = 6$ (9.8% decrease), $H = 8$ (17.1% decrease), and $H = 10$ (27.6% decrease). This pattern clearly demonstrates how exposure bias and compounding errors become increasingly problematic as action sequences extend.

In contrast, with S-AA tuning, the performance demonstrates notable improvements. When $H$ increases from 1 to 2, performance actually improves from 75.2% to 78.0%, demonstrating that our approach not only extends the generation horizon but also enhances overall task success. The performance decline as $H$ increases further is significantly more gradual with S-AA than with L-BC only. At $H = 4$, S-AA shows only a 1.3% drop from $H = 2$, compared to L-BC only's 3.2% decline over the same range. This pattern continues at $H = 6$, where S-AA's performance decreases by 7.7% from $H = 2$, while L-BC only drops by 9.2%. Even at longer horizons ($H = 8$ and $H = 10$), S-

Table 2: The results of the $Pass@N$ metric under different task suites and horizon lengths.

| | | H | Spatial | Object | Goal | Long | Average |
|---|---|---|---|---|---|---|---|
| OpenVLA | Pass@2 | 1 | 85.8±1.0% | 89.6±0.8% | 81.4±1.2% | 54.8±1.1% | 77.9±1.0% |
| | Pass@5 | 1 | 86.6±0.7% | 90.4±1.1% | 83.2±0.9% | 57.4±1.0% | 79.4±0.9% |
| Action Stream | Pass@2 | ×2 | 94.4±0.8% | 92.0±0.9% | 93.6±0.7% | 80.6±1.2% | 90.2±0.9% |
| | | ×4 | 92.8±1.0% | 92.6±0.8% | 89.2±1.1% | 76.4±0.9% | 87.8±1.0% |
| | | ×6 | 90.8±0.9% | 86.0±1.2% | 82.4±1.0% | 69.2±1.1% | 82.1±1.1% |
| | | ×8 | 82.2±1.1% | 85.4±0.7% | 72.0±1.2% | 62.4±1.0% | 75.5±1.0% |
| | | ×10 | 71.8±1.2% | 72.0±1.1% | 64.4±0.9% | 46.6±1.3% | 63.7±1.1% |
| | Pass@5 | ×2 | 98.8±0.6% | 97.6±0.7% | 95.2±0.8% | 87.4±0.9% | 94.8±0.8% |
| | | ×4 | 97.4±0.7% | 96.4±0.8% | 93.6±0.9% | 82.8±1.1% | 92.6±0.9% |
| | | ×6 | 96.6±0.8% | 95.8±0.9% | 88.2±1.0% | 76.6±1.2% | 89.3±1.0% |
| | | ×8 | 88.4±1.0% | 91.2±0.9% | 85.4±1.1% | 69.4±1.2% | 83.6±1.1% |
| | | ×10 | 82.2±1.1% | 84.6±1.0% | 83.2±0.9% | 58.2±1.3% | 77.1±1.1% |

AA maintains substantially better resilience, with relative degradation rates 13.6% and 14.9% lower than L-BC only. This clearly demonstrates that our approach effectively mitigates the compounding errors that plague longer-horizon action generation, allowing the model to maintain coherence over extended sequences.

## 4.3 ACTION STREAM SAMPLING AND DECODING ANALYSIS

Action Stream generation is essentially a form of LLM text generation, with decoding strategies significantly affecting output quality. While OpenVLA uses greedy decoding for efficiency, this approach often compromises quality and diverges from best practices in both text (Shi et al., 2024) and action generation. For robotic tasks, generation diversity is essential since multiple valid action sequences typically exist for a same state, and exploring alternatives can substantially improve task performance (Chi et al., 2023). Therefore, we investigate two alternative approaches: stochastic sampling and structured search.

**Uncovering Latent Potential with Pass@N** We employ top-$k$ stochastic decoding, where at each token generation step, the model samples from the $k$ highest-probability tokens, and this process continues autoregressively throughout the episode. We evaluate using the $Pass@N$ metric, which counts a success if at least one trajectory completes the task. Table 2 presents the results across different task suites and horizon lengths. For $H = 2$, success rates increase by 8.6%, and notably, the $Pass@2$ average performance when $H = 6$ (82.1%) surpasses the $H = 1$ baseline (77.9%). With $N = 5$, all metrics show substantial improvements, with gains reaching up to 15.4% for longer horizons. Meanwhile, OpenVLA shows minimal benefits from $Pass@N$ in single-step greedy decoding. These findings demonstrate that long-horizon generation enables better exploration of the solution space, making our action stream policy a more effective proposer. Similar to how LLMs explore the reasoning space through extended generation, our approach allows the policy to discover diverse valid action sequences within the robotic solution space.

**Exploiting the Search Space with Beam Search** The high $Pass@N$ results show our action stream policy generates multiple viable trajectories, creating a rich solution space that can be systematically explored using beam search. Beam search maintains a set of promising partial sequences at each action unit step, expands them in parallel, and selects the highest-scoring complete action stream based on cumulative log probabilities (Vijayakumar et al., 2016).

Figure 3 shows the beam search results under different beam width $B$. The results demonstrate that beam search significantly improves performance across all horizon lengths. With $B = 2$, performance at $H = 2$ substantially surpasses the baseline across all metrics. At $B = 4$ and $B = 5$, even $H = 4$ performance exceeds the baseline, demonstrating effective exploration of the solution space. Moreover, as beam width increases, the performance degradation trend with increasing horizon $H$ becomes increasingly gradual. This confirms that structured search techniques effectively exploit the rich solution space produced by our action stream policy.

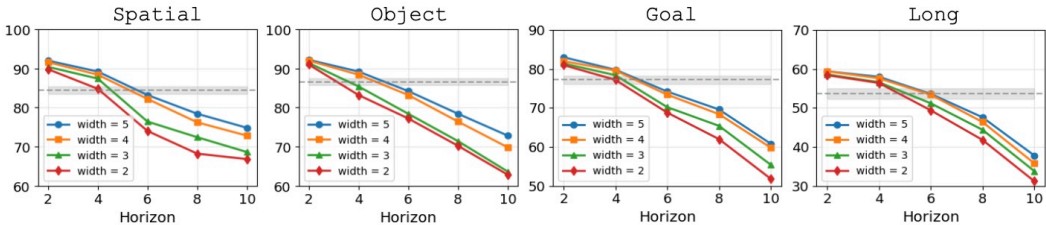

Figure 3: The beam search results under different beam width across different task suites. The gray bars represent the OpenVLA baseline performance.

## 4.4 ACTION TRAJECTORY ANALYSIS

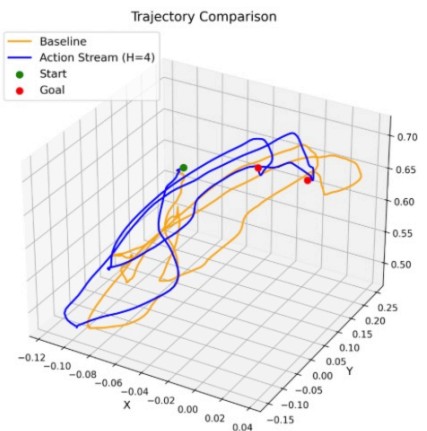

Figure 4: The trajectory visualization.

Figure 4 shows the robot end-effector movement trajectories under different horizon lengths. We observe that when the policy's generation horizon is extended, the trajectories become smoother, while step-wise prediction exhibits noticeable spikes. This phenomenon can be attributed to the fundamental difference in action generation mechanisms: step-wise prediction requires independent action prediction at each timestep under new environmental states, leading to potential inconsistencies and abrupt changes in the trajectory (Liu et al., 2024). In contrast, Action Stream performs autoregressive continuous action prediction within the same environmental context, where each action is conditioned on the preceding actions in the sequence, resulting in stronger coherence and smoother robot movements. More analysis can be found in the Appendix A.3.

## 4.5 RESULTS ON REAL ROBOT

Using an Xbox Gamepad, we teleoperate the Jaco2 arm and collect demonstrations at a frequency of 10 Hz, recording RGB images and robot states throughout each trajectory. We design four distinct tasks for evaluation: PlaceBlock, PlaceCarrot, CleanTable, and RelocateCup. We gather 50–200 demonstrations per task to form the training dataset and finetune the Action Stream using our proposed two-phase approach. Table 3 shows the results, which demonstrate consistent observations with the simulation environment. More task and training details can be found in the Appendix A.2.

Table 3: Real robot task success rates (%) with different horizon lengths.

|  | H | PlaceBlock | PlaceCarrot | CleanTable | RelocateCup | Average |
|---|---|---|---|---|---|---|
| OpenVLA | 1 | 85.0 | 90.0 | 80.0 | 75.0 | 82.5 |
| Action Stream | ×2 | 85.0 | 95.0 | 90.0 | 80.0 | **87.5** |
|  | ×3 | 80.0 | 85.0 | 85.0 | 75.0 | 81.25 |
|  | ×4 | 70.0 | 85.0 | 80.0 | 70.0 | 76.25 |
|  | ×5 | 65.0 | 75.0 | 70.0 | 65.0 | 68.75 |

## 5 RELATED WORK

### 5.1 TRANSFORMING VLM TO VLA

Recent vision-language-action (VLA) models adapt pre-trained VLMs for robotic control through two main approaches: output space unification, which integrates all modalities into a shared token space, and module grafting, which attaches specialized action prediction components to VLMs.

In the output space unification paradigm, all modalities are unified into a shared discrete token space for autoregressive modeling. Reed et al. (2022) serializes text, images, actions, and proprioception

into unified token sequences. The RT series (Brohan et al., 2022; 2024; Belkhale et al., 2024) discretizes robot actions into bins for autoregressive inference. Open X-Embodiment (O'Neill et al., 2024) standardizes action representation across diverse robots. These methods preserve the original foundation model architecture, enabling efficient end-to-end training. In the module grafting paradigm, specialized action prediction modules are attached to pre-trained VLMs. LangLfP (Lynch & Sermanet, 2020) uses a conditional variational autoencoder to generate continuous control commands. BC-Z (Jang et al., 2022) employs a VLM-based task encoder with a separate MLP network, decoupling task understanding from action generation. HybridVLA (Liu et al., 2025) integrates autoregressive and diffusion policies into a unified model. These approaches require integrating heterogeneous modules, introducing complexity in design and training.

## 5.2 Extension Policy Horizon

Extending policy horizons enables more coherent planning and reduces replanning frequency for complex sequential tasks. Long-VLA (Fan et al., 2025) allows policies to handle longer, multi-step manipulations by segmenting tasks and selectively focusing on relevant inputs. MuST(Gao et al., 2025) extends policy horizons by decomposing tasks into reusable skills and sequencing them through a progress-guided selector. Diffusion policy (Chi et al., 2023) and OFT (Kim et al., 2025) integrate action chunking decoding module to achieve multi-step action generation in parallel. However, these methods do not adopt a unified action-as-language perspective for coherent sequence generation.

## 6 Discussion and Limitation

While our work significantly advances the Action-as-Language paradigm, several limitations remain. First, action discretization remains a fundamental challenge, disrupting pose continuity and introducing quantization errors that accumulate over longer sequences, ultimately hindering precise control in high-precision tasks (Liu et al., 2025). This limitation ultimately constrains the upper performance bound of VLA models in tasks requiring fine-grained manipulation.

Additionally, Action Stream's performance degrades with longer horizons due to cumulative errors and trajectory drift. Unlike text generation, robotic actions directly interact with the physical environment and can cause irreversible changes, making each prediction error more consequential and harder to correct downstream. We provide detailed analysis of this trajectory drift phenomenon in the Appendix A.3. Moreover, Action Stream's autoregressive generation introduces computational overhead compared to single-step prediction methods, potentially causing higher latency during deployment. Future work could explore more efficient decoding strategies or hybrid approaches that balance the benefits of action stream generation with computational efficiency.

## 7 Conclusion

In this paper, we introduced the Action Stream paradigm, which advances the Action-as-Language paradigm in VLA models. Unlike traditional approaches that rely on single-step action prediction, we successfully customized LLM training and inference recipes for VLA models, overcoming the limitations of conventional methods and fully leveraging the long-sequence generation capabilities of LLMs. Through our two-phase approach consisting of offline long-horizon behavior cloning and step-wise action alignment, we enable VLA models to generate extended action sequences while effectively addressing exposure bias in long-horizon prediction. Furthermore, we successfully adapted LLM inference-time decoding techniques to the VLA domain, enabling our approach to better explore the solution space by generating multiple action trajectories and selecting the most promising ones. These inference-time enhancements unlock the potential of Action Stream, leading to significant performance improvements. This comprehensive framework substantially advances the Action-as-Language paradigm in VLA models, representing a significant step towards unified vision-language-action modeling and inspiring future work to transfer more LLM paradigms to advance VLA capabilities.

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

## A APPENDIX

### A.1 USE OF LARGE LANGUAGE MODELS

We use LLMs only for text polishing and language refinement to improve the clarity and readability of our manuscript. All core ideas, experimental designs, methodological contributions, and technical innovations presented in this work are conceived and developed entirely by human researchers. The LLMs were not involved in any aspect of the research process, including problem formulation, algorithm design, experimental setup, data analysis, or result interpretation. Their usage was strictly limited to enhancing the linguistic quality of the written content.

### A.2 REAL ROBOT EXPERIMENTS SETTINGS

**Hardware and setup.** We use a Kinova Jaco2 arm for real-world evaluation. Demonstrations are collected via Xbox Gamepad teleoperation at a control frequency of 10 Hz, recording synchronized RGB images and robot states at each step. We design four manipulation tasks: 1) **PlaceBlock**: Place the blue cube on the green plate. 2) **PlaceCarrot**: Place the carrot on the blue pan. 3) **CleanTable**: Clean the table. 4) **RelocateCup**: Pick up the orange pot and place it in the green plate. The visual observations for each task are shown in Figure 5.

**Training data.** For each task, we collect a set of human teleoperated demonstrations to construct the training dataset: 198 trajectories for PlaceBlock (average length 75 steps), 201 for PlaceCarrot (81 steps), 69 for CleanTable (58 steps), and 179 for RelocateCup (150 steps). We then finetune the Action Stream policy on these datasets using our two-phase procedure described in Section 3.

**Evaluation protocol.** For each trained policy, we perform 20 independent trials per task (80 trials in total). At the start of each trial, the robot is reset to a standardized home configuration, and the objects are placed at randomized but feasible positions on the workspace. The policy then executes until task completion or a timeout of 30 seconds.

**Success criteria.** A trial is counted as successful if the task-specific goal is satisfied: (i) the target object is fully placed inside the designated container (PlaceBlock, PlaceCarrot, RelocateCup), or (ii) the distractor object is completely removed from the designated region (CleanTable). The object must remain stable in the target region for at least one second to be considered a success. Success rates reported in Table 3 are computed as the fraction of successful trials over the total trials.

### A.3 ACTION TRAJECTORY ANALYSIS

In this section, we read trajectory data from the LIBERO simulation platform and visualize it to conduct an in-depth analysis of the robot end-effector movement trajectories. We examine two key aspects of our Action Stream paradigm: temporal coherence and error accumulation in long-sequence generation. We compare the trajectory patterns generated by our Action Stream paradigm against the baseline Openvla single-step prediction method.

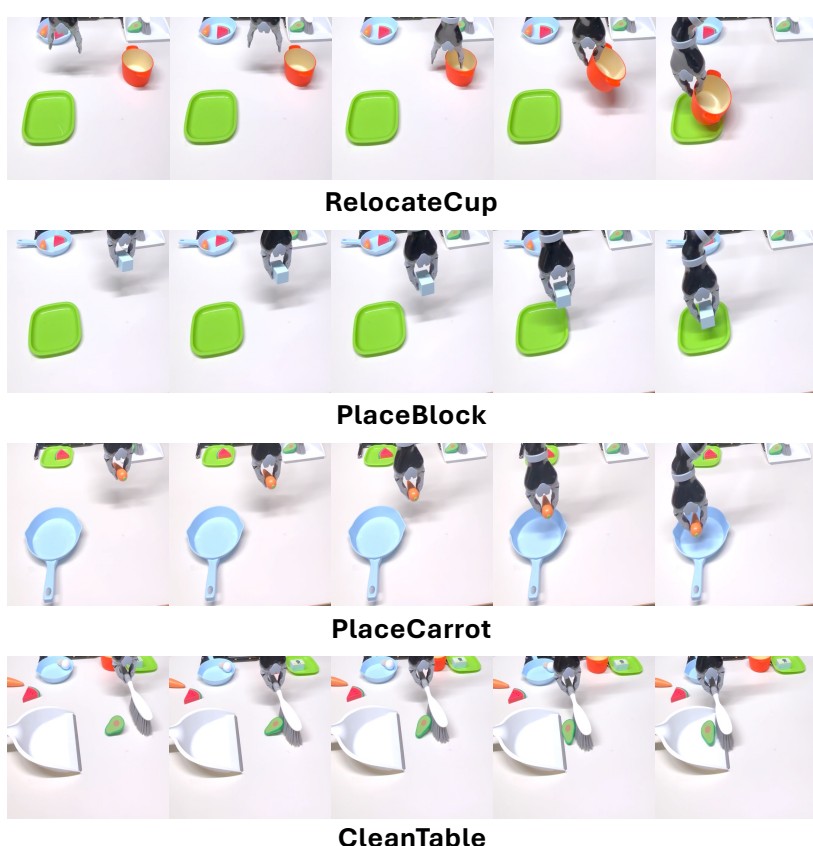

**RelocateCup**

**PlaceBlock**

**PlaceCarrot**

**CleanTable**

Figure 5: Visual observations for the four real-robot evaluation tasks.

**Temporal Coherence:** Figure 6 illustrates the end-effector trajectories for both approaches during task execution. The baseline step-wise action prediction exhibits noticeable **trajectory spikes and discontinuities (highlighted in red boxes)**, demonstrating weak temporal coherence in single-step prediction. This phenomenon can be attributed to the fundamental difference in action generation mechanisms: step-wise prediction requires independent action prediction at each timestep under new environmental states, leading to potential inconsistencies and abrupt changes in the trajectory (Liu et al., 2024). These abrupt changes in the trajectory indicate that the policy struggles to maintain smooth, consistent motion patterns when predicting actions independently at each timestep. In contrast, our Action Stream approach generates **significantly smoother trajectories with better temporal consistency**. Action Stream performs autoregressive continuous action prediction within the same environmental context, where each action is conditioned on the preceding actions in the sequence, resulting in stronger coherence and smoother robot movements. By leveraging the LLM's autoregressive generation capabilities to produce coherent multi-step action sequences, the policy maintains better continuity in the end-effector's movement patterns.

**Understanding Trajectory Drift through the Momentum Effect:** Our analysis reveals a distinctive momentum characteristic in Action Stream generation. As shown in Figure 7, while both approaches exhibit similar and overlapping trajectories in the initial stages, Action Stream exhibits a tendency to persist on pre-computed trajectories due to its autoregressive nature. Unlike single-step prediction that adapts to each new state independently, Action Stream operates in an open-loop fashion where the entire sequence of actions is conditioned solely on the initial state. This creates a form of executional momentum where the policy maintains its planned trajectory even when environmental conditions change during execution. While this momentum contributes to the smoother trajectories we observe, it also renders the policy less sensitive to real-time state changes that occur during the stream's execution, potentially leading to trajectory drift when the initial plan becomes suboptimal.

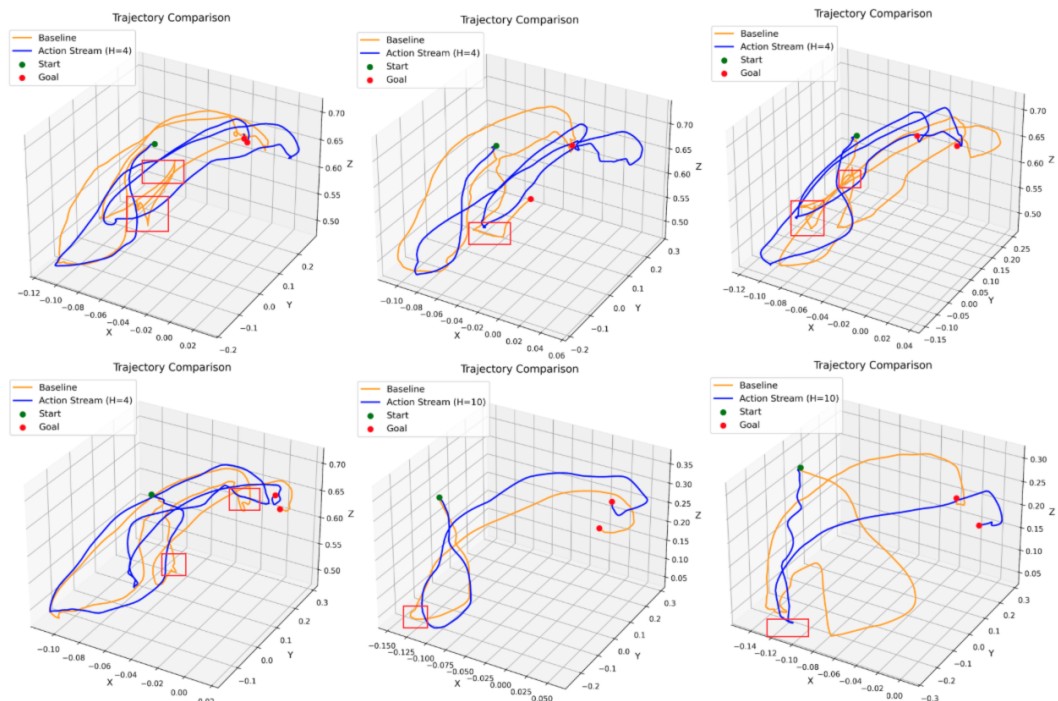

Figure 6: The action trajectory analysis results. The baseline step-wise action prediction exhibits trajectory spikes and abrupt directional changes (highlighted in red boxes), demonstrating weak temporal coherence in single-step prediction.

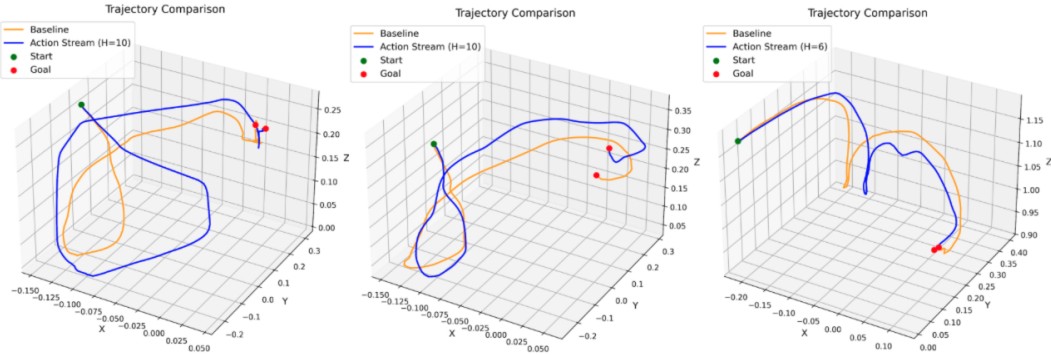

Figure 7: The action trajectory analysis results showing momentum effect. Both approaches exhibit overlapping initial trajectories, but Action Stream exhibits trajectory drift over time due to its momentum-driven autoregressive nature, persisting on pre-computed trajectories.

## A.4 TRAINING DETAILS

**Supervised Fine-Tuning (SFT) Details**  In the SFT stage, we follow the standard SFT paradigm commonly used in LLMs. We randomly sample an expert demonstration, which includes image embeddings and text token embeddings as input, and concatenate the subsequent multi-step actions as supervised output targets. The training context length is set to 512 tokens. If the concatenated multi-step action sequence does not fill the entire context length, we pad it with placeholder tokens that do not participate in the loss function computation. Figure 8 shows the SFT training details. As can be observed from the training curves, the model successfully converges with steadily decreasing loss and accuracy approaching 100%, demonstrating effective learning of the multi-step action sequences and the ability to successfully imitate long-horizon expert trajectories.

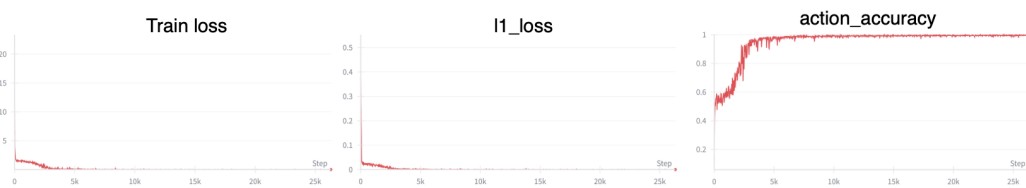

Figure 8: The SFT training details. The model successfully converges with steadily decreasing loss and accuracy approaching 100%, demonstrating effective learning of the multi-step action sequences and the ability to successfully imitate long-horizon expert trajectories.

**Step-wise action Training Details**   Since this stage requires online policy exploration and OpenVLA's setup does not support batch inference, we use a batch size of 1 during this phase. We sample a state from expert demonstrations and obtain the subsequent H-step action sequence, while also allowing the policy to explore for 10 steps (generating 10×8=80 tokens, consisting of 7 action dimensions plus 1 separator token per step). We then compute preference rewards for each step to identify the first error. DPO loss is applied only to the first error step. For training stability, we maintain a 9:1 ratio between DPO loss and SFT loss. If no first error is identified, we perform SFT fine-tuning using the expert demonstration.

During the training process, we monitor two key metrics: the expected action reward (representing expert demonstrations) and the policy's online exploration reward. Figure 9 illustrates the convergence curves for these two rewards, where "winner" represents the expected expert actions and "loser" represents the policy-generated actions. As training progresses, both rewards show an upward trend, indicating that the policy increasingly favors expert actions while simultaneously generating actions that closely align with expert behavior. This dual improvement demonstrates that our Step-wise Action Alignment effectively guides the policy to prefer expert demonstrations while enhancing the quality of its own action generation, leading to a virtuous cycle where policy-generated actions become increasingly expert-like and thus receive higher preference scores.

We select the optimal checkpoint for evaluation based on the convergence behavior of both expert and policy rewards. Specifically, we monitor the reward gap between expert demonstrations (winner) and policy-generated actions (loser) throughout training. The checkpoint is selected when: (1) the reward gap between expert and policy actions becomes minimal, indicating that policy-generated actions closely match expert quality, and (2) both reward curves exhibit stability without significant fluctuations, suggesting convergence. This selection strategy ensures that we evaluate the model at its optimal performance point where the policy has learned to generate expert-like actions consistently.

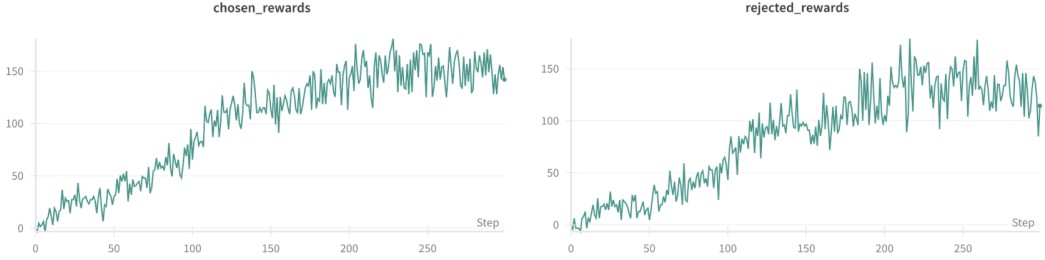

Figure 9: The DPO training details showing winner and loser reward changes. Winner represents expert trajectory actions while loser represents policy exploration actions. The upward trend of both rewards indicates that policy exploration actions are increasingly aligning with expert actions.

## A.5   LIBERO SIMULATION BENCHMARK SETTINGS

LIBERO (Liu et al., 2023) is a recent benchmark for embodied robot learning, designed to evaluate generalization across diverse manipulation scenarios. It consists of four task suites, each contain-

Figure 10: Visual observations for the four LIBERO evaluation task suites. We show episodes from 4 subtasks to demonstrate the diversity and complexity of the evaluation scenarios.

ing 10 subtasks with 50 human-teleoperated demonstrations per task. The suites differ in object diversity, spatial variation, and goal specification, thereby testing complementary aspects of policy robustness:

- **LIBERO-Spatial**: Same set of objects, but placed in different spatial layouts. This suite evaluates the ability to generalize across spatial configurations and adapt to changes in object positions and relative distances.
- **LIBERO-Object**: Same spatial layouts, but involving different objects. This suite tests object recognition and manipulation transfer when new but semantically similar objects are introduced.
- **LIBERO-Goal**: Same objects and layouts, but with varying goals or instructions. This suite emphasizes understanding task semantics and aligning actions with different high-level task objectives.
- **LIBERO-Long**: Also known as LIBERO-10, containing long-horizon tasks with multiple objects, diverse layouts, and compound goals. This suite is the most challenging, as it requires coherent multi-step planning and strong temporal consistency to succeed.

Together, these four suites provide a comprehensive evaluation: LIBERO-Spatial and LIBERO-Object measure generalization to environment or object shifts, LIBERO-Goal probes semantic grounding, and LIBERO-Long stresses long-horizon reasoning and error accumulation. Figure 10 shows example episodes from each suite, illustrating their diversity and complexity.

