# OpenReview forum: "From Language to Action Streams: Bridging LLM Autoregression for Long-Horizon Robot Action Prediction"
_ICLR.cc/2026/Conference — ICLR 2026 Conference Withdrawn Submission_

### Official Review · Reviewer_kePK · 2025-10-27

**Soundness:** 2
**Presentation:** 3
**Contribution:** 3
**Rating:** 6
**Confidence:** 4

**Summary:**

This paper proposes a simple extension to the Vision-Language Action model paradigm where the VLA is tasked with producing additional steps into the future when predicting an action at the current timestep. The training of this model, built on Llama 2, introduces two training methods: long horizon behaviour cloning and step-wise action alignment. The first step of training enables multi-step action prediction, while the second step of training attempts to limit bias during inference. Results on simulated and real-world robots seem promising.

**Strengths:**

1. A simple but quality method that increases performance without too much additional overhead.
2. The paper is well-written and results across simulation and real-world settings are a bonus.

**Weaknesses:**

1. I think there are missing baselines (RT-1-X, RT-2-X, Octo, for example) that could make the work even stronger. Showing that the two step approach for generating multiple steps of actions improves performance across a broad range of models would make this work much stronger. I think this would fit within the paper very well, because the authors don't claim SOTA for their specific algorithm, but showing that their proposed two step training method improves performance across additional VLA models would show that this could be a fruitful paradigm to use as the standard for training any future VLA models. I think validating the additional action steps are a benefit across more models should be tested in simulation.
2. I think some discussion & comparison of world models would be incredibly valuable to the reader for full context of different methods in this space. In my view, these VLA models boil down to world models (from the RL literature i.e. Dreamer V3 or TD-MPC-2) where the VLA models have a similar notion of how their actions will impact the environment/task that they are interacting with.
3. For each of the tables, some more information is required to ensure that full context of what the table contains can be gathered. For example, in Table 1, what does the bolding of results mean? What is H? What are the labels of Spatial, Object, etc? I think making the descriptions of these tables contain this information would be a large help to the reader.

**Questions:**

1. How does the VLA paradigm relate to the world model paradigm in RL?
2. The proposed method doesn't leverage any auxiliary objectives for training (i.e. predicting the next state, predicting the next action). Do the authors think the integration of these objectives could enable predictions even further into the future? Would the bias mitigation strategy be needed if these objectives were added?
3. Results in simulation seem to show means and standard deviations, how many seeds were used to generate these results?

---

### Official Review · Reviewer_ep9u · 2025-10-31

**Soundness:** 3
**Presentation:** 2
**Contribution:** 2
**Rating:** 2
**Confidence:** 4

**Summary:**

This paper introduces the Action Stream paradigm for Vision-Language-Action (VLA) models, customizing OpenVLA training and inference to enable long-horizon action sequence generation in robotics. The approach involves a two-phase training method: Long-horizon Behavior Cloning (L-BC) for imitating expert streams and Step-wise Action Alignment (S-AA) to mitigate exposure bias. Extensive evaluations on simulation benchmarks and real-world setups demonstrate improved task performance with extended generation horizons.

**Strengths:**

- Innovative Framework for Long-Horizon Generation​​. The Action Stream leverages the inherent coherence of autoregressive models, allowing policies to generate smooth, multi-step actions from a single observation. The paradigm shift from step-wise to stream-wise prediction reduces temporal inconsistencies, as visualized in trajectory analyses.

- ​​Robust Training Methodology with Error Mitigation​​. L-BC provides a foundation for imitating expert demonstrations, while S-AA introduces online exploration and alignment via Direct Preference Optimization (DPO), pinpointing and correcting the first deviation in action sequences. This reduces compounding errors and enhances policy reliability, as evidenced by improved success rates across horizons.

**Weaknesses:**

- The article's novelty is limited. The authors merely added multi-step action prediction on top of OpenVLA, which does not significantly differ from the original VLA architecture.

- The paper fails to address key VLA limitations, particularly limited generalization, especially in text generalization.

- The article only compares with one baseline OpenVLA, and the results are clearly not state-of-the-art on LIBERO benchmarks.

- As horizon H increases, Action Stream performance declines, indicating severe cumulative prediction error. This may stem from limited data scale, suggesting VLA has not truly learned underlying action space patterns.

**Questions:**

The same as weekness

---

### Official Review · Reviewer_TZL1 · 2025-11-01

**Soundness:** 3
**Presentation:** 3
**Contribution:** 1
**Rating:** 2
**Confidence:** 3

**Summary:**

The authors present Action Streams, a VLA training paradigm with two components: a long-horizon behavior cloning phase (L-BC), which trains a VLA model to extend its action generation horizon, and a step-wise action alignment step phase (S-AA), which accounts for error accumulation in the long-horizon action output. Action Streams is built on top of OpenVLA. L-BC is implemented by creating long-horizon action streams from the current data with concatenation, then performing supervised fine-tuning. S-AA is implemented by performing Direct Preference Optimization, using the supervised fine-tuned policy as the expert, and formulating the loss based on when the most likely actions between the expert and reference policy diverge. Experiments are conducted on a real world set up, and on Libero to analyze and ablate Action Streams.

**Strengths:**

- Action Streams is well motivated and clearly detailed.
- The Action Streams results on long horizons are interesting and promising, as shown in Table 1, and show that S-AA has the potential to combat error accumulation on longer horizons.

**Weaknesses:**

- The novelty of Action Streams is unclear. How is L-BC different from instruction tuning? Is the S-AA component the main contribution?
- The experiments lack other long-horizon VLA methods (such as those mentioned in the related works section) as baselines.

**Questions:**

- Other methods such as $\pi_{0}$ [1] generate actions over multiple timesteps and perform temporal aggregation to generate the final action. How does Action Streams improve over this method for long-horizon action generation?
- Is the loss in equation 8 novel?
- Did OpenVLA and Action Streams see the same data in Table 2?
[1] $\pi_0$: A Vision-Language-Action Flow Model for General Robot Control. Black et al. arxiv preprint arXiv:2410.24164 2024.

---

### Official Review · Reviewer_KQrT · 2025-11-01

**Soundness:** 1
**Presentation:** 2
**Contribution:** 1
**Rating:** 0
**Confidence:** 5

**Summary:**

This paper proposes the Action Stream paradigm for Vision-Language-Action (VLA) models, aiming to extend the policy generation horizon by enabling LLMs to generate multi-step action sequences autoregressively. The authors introduce a two-stage training scheme: (1) Long-Horizon Behavior Cloning (L-BC) to learn coherent action streams, and (2) Step-wise Action Alignment (S-AA) using Direct Preference Optimization to mitigate exposure bias. Experiments are conducted on the LIBERO benchmark and a Kinova Jaco2 robot.

**Strengths:**

1. The incorporation of DPO-based alignment is interesting.
2. The evaluation is done in both simulation and real-world setups.

**Weaknesses:**

1. The proposed “multi-step prediction” largely corresponds to modifying OpenVLA to generate multiple actions in one forward pass, which is something prior work[5] already implemented. The contribution is therefore incremental.
2. The method is built based on OpenVLA but OpenVLA is no longer the SOTA. Most of the SOTA VLAs such as Pi[3,4] do multiple step prediction (e.g. pi0[3] do a 50 step prediction), combined with receding horizon control/ temporal assemble, they can achieve much better performance (e.g. pi0.5[4] has an average of 0.97 success rate on libero tasks with multiple steps prediction and execution [2]). The proposed method yields lower task success at longer horizons (Tables 1 and 3). The core idea of autoregressive long-horizon prediction is neither novel nor shown to improve real performance. The incremental beam-search decoding further limits its originality.
3. The contribution of the proposed method is questionable.  It seems the authors want to do long horizon prediction but the shown performance actually decreases with longer horizon. Then what’s the argument of doing long horizon with the proposed method? And what’s the contribution of the proposed method? Furthermore, the paper didn't compare with other SOTA VLAs doing long horizon predictions as aforementioned.
4. “current implementations underutilize the LLM’s full generation potential, confining action prediction to fixed-length, single-step token sequences and limiting the policy’s generation horizon. ” This is not true, for example, FAST[1] can tokenize a sequence of actions into one token, there are also other action tokenizers works that predicts varying length multi step token sequences.
5. What is the expert data at the online exploration process? If it's the same training dataset, this doesn’t make sense, because the action can be multi-mode and deviating from the expert doesn’t mean failure at test time.

[1] Pertsch, Karl, et al. "Fast: Efficient action tokenization for vision-language-action models." arXiv preprint arXiv:2501.09747 (2025).

[2]  Zhou, Xueyang, et al. "LIBERO-PRO: Towards Robust and Fair Evaluation of Vision-Language-Action Models Beyond Memorization." arXiv preprint arXiv:2510.03827 (2025).

[3] Black, Kevin, et al. "$\pi_0 $: A Vision-Language-Action Flow Model for General Robot Control." arXiv preprint arXiv:2410.24164 (2024).

[4] Intelligence, Physical, et al. "π0. 5: a vision-language-action model with open-world generalization, 2025." URL https://arxiv. org/abs/2504.16054 1.2: 3.

[5]  Huang, Huang, et al. "Otter: A vision-language-action model with text-aware visual feature extraction." arXiv preprint arXiv:2503.03734 (2025).

**Questions:**

see weakness 5.

---

### Note · Authors · 2025-11-14

**Comment:**

Thank you for all the valuable reviews. We will continue to refine our work.

**Withdrawal Confirmation:**

I have read and agree with the venue's withdrawal policy on behalf of myself and my co-authors.